# Bio-Inspired Internet of Things: Current Status, Benefits, Challenges, and Future Directions

**DOI:** 10.3390/biomimetics8040373

**Published:** 2023-08-17

**Authors:** Abdullah Alabdulatif, Navod Neranjan Thilakarathne

**Affiliations:** 1Department of Computer, College of Sciences and Arts in Al-Rass, Qassim University, Al-Rass 720223, Saudi Arabia; a.alabdulatif@qu.edu.sa; 2Department of ICT, Faculty of Technology, University of Colombo, Colombo 00700, Sri Lanka

**Keywords:** IoT, Internet of Things, bio-inspired IoT, bio-inspired computation, bio-inspired algorithms, artificial intelligence, cyber-security, optimization, wireless sensor network

## Abstract

There is no doubt that the involvement of the Internet of Things (IoT) in our daily lives has changed the way we live and interact as a global community, as IoT enables intercommunication of digital objects around us, creating a pervasive environment. As of now, this IoT is found in almost every domain that is vital for human survival, such as agriculture, medical care, transportation, the military, and so on. Day by day, various IoT solutions are introduced to the market by manufacturers towards making our life easier and more comfortable. On the other hand, even though IoT now holds a key place in our lives, the IoT ecosystem has various limitations in efficiency, scalability, and adaptability. As such, biomimicry, which involves imitating the systems found in nature within human-made systems, appeared to be a potential remedy to overcome such challenges pertaining to IoT, which can also be referred to as bio-inspired IoT. In the simplest terms, bio-inspired IoT combines nature-inspired principles and IoT to create more efficient and adaptive IoT solutions, that can overcome most of the inherent challenges pertaining to traditional IoT. It is based on the idea that nature has already solved many challenging problems and that, by studying and mimicking biological systems, we might develop better IoT systems. As of now, this concept of bio-inspired IoT is applied to various fields such as medical care, transportation, cyber-security, agriculture, and so on. However, it is noted that only a few studies have been carried out on this new concept, explaining how these bio-inspired concepts are integrated with IoT. Thus, to fill in the gap, in this study, we provide a brief review of bio-inspired IoT, highlighting how it came into play, its ecosystem, its latest status, benefits, challenges, and future directions.

## 1. Introduction

The proliferation of Information and Communication Technologies (ICTs) early in the 21st century has led to a revolution in our daily lives in which we are able to perform many tasks that were considered difficult with the use of available technologies at that time [1]. Progressively, this ICT has given birth to many advanced technologies, of which IoT can be considered as one such major technology [2,3,4,5]. Over the years, this IoT has been affiliated with many of the domains that are deemed important for human survival, such as agriculture, manufacturing, transportation, and healthcare [6,7,8,9]. Undoubtedly, as of now, this IoT can be considered a major technology that is an essential part of our daily lives. Day by day, IoT solutions are becoming more closely integrated into our daily lives, and such IoT solutions are being introduced to the market at a rapid rate. In general, the IoT has the capability to connect digital devices/objects all around the world and allow them to communicate with each other, creating a larger ubiquitous network [10,11,12].

Overall, the IoT is an ecosystem of connected devices that communicate with each other and exchange data. The integration of the IoT and the worldwide web has led to the creation of new kinds of services and applications in different domains, such as real-time surveillance, plant disease diagnosis, remote healthcare, real-time disease diagnosis, precision agriculture, air quality monitoring, and so on [1,2,3,4,5,12,13,14,15,16]. In general, IoT facilitates us to connect our digital belongings to the worldwide web, anytime, anywhere in the world [17,18,19,20,21], as depicted in Figure 1.

Day by day, various IoT solutions are being introduced to the market owing to the development of underlying infrastructure (e.g., communication facilities, computing facilities) and the increasing technology competency of people. These ever-growing IoT solutions are employed in a variety of domains and offer convenient facilities to the stakeholders engaged in these domains. According to the latest statistics, it is predicted that there will be over 100 billion IoT devices by 2025 [5,6,7,8,9,10]. It is also evident that the IoT market size is expected to reach over USD 140 billion by 2030 from USD 12 billion in 2020 [5,6,7,8,9,10]. Overall, the IoT is transforming industries and revolutionizing the way we live and work. However, even though IoT is a fast-growing technology, it is still in its infancy, and it will take years to become a stable technology [21,22,23,24,25]. As such, there are several drawbacks and challenges associated with the incorporation and implementation of the IoT (depicted in Figure 2), which are categorized and highlighted further in the following.

Security and privacy

Since IoT devices are always connected to the Internet, they can be vulnerable to hacking, unauthorized access, and data breaches [10]. Further, weak security measures and insufficient encryption can expose sensitive information, leading to privacy violations and potential risks to individuals and organizations [26,27,28].

Interoperability issues

The IoT ecosystem involves a wide range of devices, protocols, and platforms from various manufacturers [4]. Achieving seamless interoperability and compatibility among these different components can be challenging whereas lack of standardized communication protocols and fragmentation within the IoT industry can hinder device connectivity and hinder the integration of IoT systems [4,5].

Scalability and complexity

As the number of IoT devices increases, managing and scaling the infrastructure becomes more complex [4,5]. Connecting and coordinating a large number of devices, managing data flows, and ensuring efficient data processing and analysis require robust systems and architectures. Thus, scaling up IoT deployments can be challenging, particularly when dealing with legacy systems and integrating with existing IT infrastructure [3].

Power and energy requirements

Many IoT devices are small, battery-powered devices designed to operate for extended periods without human intervention. However, maintaining constant connectivity, data transmission, and processing can drain device batteries quickly [10,11,12,13]. Thus, energy efficiency becomes a crucial factor, especially for IoT deployments that involve a massive number of devices or remote locations (e.g., in agricultural lands, mines etc.) where the power supply is limited.

Data overload and management

IoT generates vast amounts of data from multiple sources and sensors, where effectively managing, storing, analyzing, and deriving actionable insights from these data can be a daunting task [10,11,12,13]. Thus, organizations need robust data management strategies, including data storage, processing, and analytics capabilities, to make sense of the information generated by IoT devices.

Reliability and downtime

IoT devices heavily rely on network connectivity and uninterrupted Internet access [4,5]. Network outages, connectivity issues, or server downtime can impact the functionality of IoT systems. Hence, dependence on stable network infrastructure and the potential for single points of failure pose reliability challenges for IoT deployments [4,5].

Addressing these inherent drawbacks of IoT requires concerted efforts from manufacturers, policymakers, and technology experts, and it will take years to tackle all these drawbacks. As IoT continues to evolve, there is an ongoing need to prioritize security, standardization, data management, and privacy considerations to mitigate these challenges. On the whole, all the aforementioned drawbacks can be mainly apportioned into scalability, efficiency, security, and adaptability challenges. As a remedy to overcome such challenges, bio-inspired IoT emerged as a new paradigm that combines the principles of biology with IoT.

In general, these bio-inspired solutions, also known as biomimetic or nature-inspired solutions, refer to the development of technologies, strategies, and designs that draw inspiration from biological systems found in nature [3,4]. These solutions aim to solve complex problems by emulating or adapting principles, structures, processes, and behaviors observed in living organisms. Incorporating such solutions with IoT leads to the creation of innovative and efficient solutions enabling IoT to overcome its inherent drawbacks [1,2,3,4]. According to the literature, in the area of bio-inspired IoT, little research has been carried out and no reviews have synthesized the latest knowledge pertaining to the subject area owing to the novelty of the field. Thus, motivated by the synthesis of the latest knowledge pertaining to bio-inspired IoT, the following describes the key contributions of the study.

Following the Introduction, we provide a brief review of what is meant by bio-inspiration, bio-inspired IoT, and the benefits and applications of bio-inspired IoT.The latest status of bio-inspired IoT is presented with use cases (highlighting domains that they are used in).A summarization of the state of the art is presented, providing a summary of recent research and survey studies followed by a brief comparison of IoT and bio-inspired IoT.To make this a comprehensive review, challenges and anticipated future directions pertaining to the bio-inspired IoT are also highlighted.

To carry out the review, we have followed a systematic approach in which the initial step encompasses the identification of key terms relevant to the topic, “bio-inspired Internet of Things”, “IoT”, “bio-inspired algorithms”, and “IoT algorithms”. To effectively retrieve pertinent scholarly content, we have searched through the IEEE Xplore, ACM Digital Library, PubMed, Google Scholar, Scopus, and Web of Science scholarly databases. Upon executing the search queries, the next phase involves the meticulous screening of search results. Titles and abstracts of retrieved articles are carefully reviewed to gauge their relevance. To ensure the inclusion of high-quality and recent literature, a set of inclusion and exclusion criteria is established where we have chosen scholarly articles published from 2010 onwards. However, we have also referred to the state of the art in 1998, 1999, and 2001, owing to the relevance of the subject for our review (e.g., swarm intelligence). 

The remainder of the study is organized in the following order. Following the Introduction, the second section discusses bio-inspiration, bio-inspired solutions, and bio-inspired IoT. Afterwards, the third section discusses the ecosystem of bio-inspired IoT, highlighting the architecture of bio-inspired IoT and bio-inspired algorithms. Then, the fourth section summarizes the state of the art. Challenges pertaining to the bio-inspired IoT are highlighted in Section 5 and current status and future directions are highlighted in Section 6 and, finally, the study concludes with the conclusions derived from the study. The Table 1 highlights the acronyms used in the study.

## 2. Bio-Inspired IoT

With the boom of ICTs happening early in the 21st century, researchers and academics began drawing inspiration from biological solutions when developing applications and systems that are dynamic, real-time, and resilient. They have found that biological solutions are great for tackling real-world complicated engineering challenges because of their flexibility, robustness, and resilience to handle failure [3]. According to the available literature, the field of bio-inspired study can be apportioned into three primary fields [27,28,29,30], as shown in Figure 3.

Bio-inspired computing.Bio-inspired systems.Bio-inspired networking.

Bio-inspired computing focuses on optimization and efficient computing and represents a class of algorithms that focus on efficient computing, whereas bio-inspired systems constitute a class of system architectures for massively distributed and collaborative systems [25,29]. On the other hand, bio-inspired networking is a set of approaches that are used for scabble and efficient networking under uncertain, volatile conditions [7,8,29]. Overall, in order for a biological idea to be applied, it needs to be comparable to an issue that exists in the actual world. It should be possible to put into action, and the procedure for adapting it to a system in the actual world should be articulated in a way that is obvious and concise. In this regard, the next section briefly discusses biomimicry, or bio-inspiration, and then the bio-inspired IoT solutions and how they are formed.

### 2.1. What Is Biomimicry?

The concept of biomimicry derives its name from the combination of two Greek words, “bios”, meaning life, and “mimesis”, meaning imitation [26]. Essentially, it involves imitating the systems found in nature within human-made systems. Biomimicry encompasses innovation inspired by nature, where engineers, knowledgeable about mechanics and dynamic flow systems in industrial processes, collaborate with biologists who possess expertise in the mechanics and dynamic flow characteristics of living processes [27,28,29,30].

Throughout human existence, nature has provided solutions to numerous complex problems that engineers continue to face. By exploring nature all around us, we can discover elegant and environmentally friendly solutions that do not contribute to further degradation. Nature avoids the use of unnecessary toxic chemicals and manages waste effectively, either through compensation or efficient cleanup. For instance, Leonardo da Vinci conceptualized designs for helicopters and parachutes by studying natural systems, and the Wright brothers examined birds to develop their first flying machine, which paved the way for the development of modern aircraft [26,27,28,29,30]. This clearly indicates how nature has already solved many existing problems and how it holds the answers for many of the challenges that humankind is currently struggling with.

Overall, nature operates in a simple and sustainable manner. By emulating natural engineering, individuals can strive for technological advancements while living harmoniously with nature and maintaining a balanced and equitable lifestyle. This approach stands in contrast to current systems that disrupt harmony, pose risks to health through toxic elements, and strain ecosystem services. The emulation of nature’s engineering systems operates on three levels, according to [26].

Emulating the form and function of natural processes.Emulating the way nature produces (engineers) biological components.Examining and understanding how nature deals with all aspects of waste and regeneration through closed system thinking.

Starting from 1998, biomimicry has facilitated collaborations between engineering firms and scientists to create eco-friendly and efficient technology through redesign and construction. Such example designs include modern automobiles, fighter aircrafts, bullet trains, underwater submarines, etc. [26]. Overall, these nature-inspired solutions have now become an essential part of our daily lives.

Having provided a brief overview of what is meant by bio-inspiration, the next section discusses such bio-inspired solutions in more detail.

### 2.2. Bio-Inspired Solutions

As aforementioned, the concept of bio-inspiration has inspired many domains to create/implement novel solutions that can overcome many challenges that are otherwise hard to overcome with traditional technologies. The process of designing bio-inspired solutions can be apportioned into five main stages, as shown in Figure 4 [25,26,31]. It involves starting from understanding the problem to be solved (problem), abstracting the problem to be solved (reframe), exactly identifying the problem and the solution (identify), analysis of biological principles and abstraction to identify analogies, and, finally, applying bio-inspired solutions to solve the problem (apply). For a better understanding, the following describes some of the remarkable bio-inspired solutions designed in recent years inspired by the biological world.

Robotics and locomotion

Researchers study animal locomotion to create robots and vehicles that can move more efficiently and adapt to various terrains [2,3]. Such examples include robots inspired by insects, birds, or marine creatures and the design of the high-speed bullet trains in Japan, which was inspired by the beak shape and aerodynamic efficiency of the kingfisher [26], where the streamlined nose design reduces noise and enhances speed, making it an iconic bio-inspired solution in transportation. Another example is swarm robotics which involves the coordination and cooperation of multiple robots to achieve collective behavior [6,20,26]. Inspired by the behavior of social insects like ants or bees, swarm robotics has applications in various fields, including environmental monitoring, search and rescue, and industrial automation. Further, the flying patterns and behaviors of birds and insects have inspired the development of autonomous drone systems where bio-inspired algorithms enable drones to navigate through complex environments, adapt to obstacles, and mimic swarming behavior for collaborative tasks [25,26].

Materials and structures

Natural materials such as spider silk, lotus leaves, or seashells possess unique properties that can be replicated for various applications. For instance, biomimetic materials can be used to create self-cleaning surfaces, lightweight and strong structures, or flexible and stretchable materials [2,3,31].

Energy and sustainability

Natural systems often exhibit efficient energy conversion and storage mechanisms. By studying photosynthesis, researchers have developed solar cells inspired by plant structures. Additionally, swarm intelligence algorithms, inspired by the behavior of social insect colonies can optimize energy consumption and resource allocation [25,26,27,28]. 

Medicine and healthcare

Nature provides numerous examples of highly efficient and adaptive biological systems that inspire medical advancements. Biomimicry can be applied to develop new drug delivery systems, medical implants, tissue engineering techniques, or prosthetics [24,25,26]. In recent times, prosthetic limbs and robotic exoskeletons have been developed to mimic the movement and functionality of natural human limbs. These bio-inspired solutions aim to restore mobility and enhance the quality of life for individuals with limb impairments [26].

Optimization and algorithms

Bio-inspired algorithms, such as Genetic Algorithms (GAs) and Ant Colony Optimization (ACO), have inspired computational algorithms that solve complex optimization problems. Overall, these algorithms mimic the processes of evolution, cooperation, and adaptation found in nature [24,25,26,27,28].

Figure 5 depicts several bio-inspired solutions that have been designed in recent years; solar cells inspired by plants photosynthesis procedures and robots inspired by the moving pattern of insects.

Having discussed what is meant by bio-inspiration and bio-inspired solutions, the next section thoroughly discusses bio-inspired IoT.

### 2.3. What Is Bio-Inspired IoT?

Bio-inspired IoT is a concept that combines the principles of biology and IoT to create a more efficient and adaptive system. It is based on the idea that nature has already solved many complex problems, and by studying and mimicking biological systems, we can develop better IoT systems to enhance the functionality of such IoT systems toward improving security, scalability, adaptability, efficiency, and so on [1,2,3,4,5,32,33,34,35,36]. Overall, bio-inspired IoT can be defined as the use of bio-inspired computing techniques and algorithms to develop creative/novel IoT systems that can overcome the inherent challenges of IoT, such as resource constraints, low efficiency, and so on [17,18,19,20,37,38,39,40]. As of now, this bio-inspired IoT is applied to various fields such as agriculture, healthcare, transportation, smart cities, and so on. 

According to the state of the art, bio-inspired IoT has several benefits over traditional IoT systems, such as:Scalability

Bio-inspired IoT systems can scale easily due to the decentralized nature of biological systems. They can handle a large number of devices and data without compromising efficiency [23,24,25]. In this regard, bio-inspired IoT algorithms, such as swarm intelligence [9,20], enable efficient scalability in IoT systems by leveraging collective intelligence and self-organization, allowing devices to interact and cooperate locally. On the other hand, this scalability feature enables bio-inspired IoT systems to handle a growing number of devices, interactions, and data, without lacking efficiency and performance.

Efficiency

Bio-inspired IoT systems are energy-efficient as they can adapt to changing environmental conditions and optimize their operations accordingly. In general, they optimize energy consumption, adjust resource usage based on demand, and enable energy-harvesting techniques [39]. This focus on energy efficiency leads to sustainability, longer device battery life, and reduced environmental impact [23,24,25].

Adaptability

Bio-inspired IoT systems can adapt to changing conditions and learn from their environment. They can dynamically adjust to changing environmental conditions, network configurations, and resource availability. This adaptability allows bio-inspired IoT systems to handle uncertainties, fluctuations, and unexpected events more effectively as opposed to traditional IoT [5,26].

Resilience and autonomous behavior

Biological systems have evolved to be resilient to external disturbances, and bio-inspired IoT systems can inherit these properties. They can recover from failures and continue to operate efficiently under adverse circumstances [26,27,28,29,30]. On the other hand, bio-inspired IoT systems can incorporate Artificial Intelligence (AI) and adaptive behavior. By mimicking biological learning processes, they can adapt and improve their performance over time. With the involvement of AI, bio-inspired IoT systems can learn from data, adjust their parameters, and make intelligent decisions, enhancing their autonomy and ability to handle complex tasks [9,16].

Bio-inspired IoT has a variety of applications across various domains, as shown in Figure 6, which we have categorized according to the state of the art. In the following, we briefly discuss these domains.

Agriculture

Bio-inspired IoT solutions have been implemented in agriculture to improve crop yields, optimize resource usage, and monitor environmental conditions [16,30,31,32,33]. For instance, using sensors inspired by bee vision, IoT devices can monitor plant health by analyzing the reflection of light from leaves. This helps farmers make informed decisions about irrigation, fertilization, and pest control.

Healthcare

Bio-inspired IoT devices are being used for remote health monitoring and personalized medicine. For example, wearable devices inspired by the structure and functionality of human skin can continuously monitor vital signs, detect abnormalities, and transmit data to healthcare providers in real time [17,18,19,20,21]. Nonetheless, bio-inspired IoT can be used to monitor patients’ vital signs and alert healthcare professionals in case of emergencies. It can also be used to track the spread of diseases and monitor outbreaks.

Transportation

Bio-inspired IoT can be used to optimize traffic flow and reduce congestion. It can also be used to monitor vehicle performance and reduce emissions [25,26,27,28].

Environmental monitoring

Bio-inspired IoT systems can help monitor and manage environmental conditions. For instance, IoT sensors inspired by the sensory capabilities of birds can detect air pollution, monitor noise levels, or analyze water quality in real time, contributing to early warning systems and environmental conservation efforts [18,25,26].

Energy management

Inspired by the behavior of social insect colonies, bio-inspired IoT systems can optimize energy management in smart grids or buildings [25,26,27,28]. By coordinating the energy consumption and production of multiple IoT devices, these systems can improve efficiency, balance energy loads, and enable better integration of renewable energy sources.

Structural health monitoring

Bio-inspired IoT solutions can be used to monitor the structural health of buildings, bridges, and other infrastructure [3,4,5]. By mimicking the sensory capabilities of animals, such as bats or dolphins, IoT sensors can detect and analyze vibrations, acoustic signals, or changes in electromagnetic fields to identify potential structural issues or defects.

Wildlife conservation

IoT devices equipped with bio-inspired sensors and communication mechanisms have been deployed to monitor wildlife populations and protect endangered species. For example, researchers have developed bio-inspired camera traps that mimic the visual and auditory cues of prey to attract and photograph elusive or endangered animals [15,25,26].

Military

Bio-inspired IoT in the military involves applying biological principles to design adaptable, resilient, energy-efficient IoT systems capable of self-organization, enhanced sensing, swarm intelligence, stealth, and biometric security (e.g., swarm drones for reconnaissance, biochemical sensing, and energy-efficient sensors) [25,26]. It aims to leverage nature-inspired strategies for efficient data processing, communication, and environmental adaptation, enabling improved situational awareness, surveillance, reconnaissance, and mission success.

Smart cities

Bio-inspired IoT can enhance the efficiency of urban infrastructure and services. For instance, researchers have looked at the collective foraging behavior of ants to develop efficient routing algorithms for garbage collection and transportation networks [25,26]. By applying these principles, IoT devices can optimize routes, reduce congestion, and minimize energy consumption [3,25,26,27,28]. On the other hand, bio-inspired IoT is used to monitor air and water quality, detect and prevent natural disasters, and improve energy efficiency in a typical smart city.

## 3. The Bio-Inspired IoT Ecosystem

The IoT is an ecosystem that comprises various enabling technologies, of which IoT constitutes the core. The remaining technologies that constitute an IoT ecosystem include cloud computing, fog computing, edge computing, AI, underlying communication technologies, and so on. Overall, the typical architecture of an IoT solution comprises three or four layers [32,33]. The three-layer architecture comprises a physical/sensing layer, network/communication layer, and the application layer whereas the four-layer architecture comprises an intermediate data-processing layer in between the network and the application layer [1,3,4]. On the other hand, when it comes to bio-inspired IoT, the pervasive ecosystem comprises underlying bio-inspired algorithms/techniques that hold a key place in the bio-inspired IoT ecosystem. The architecture of bio-inspired IoT systems varies depending on the specific application and the level of bio-inspiration incorporated. However, there are certain common architectural elements that are often present in bio-inspired IoT systems. To provide a better understanding of the bio-inspired IoT ecosystem, the next section discusses further the architecture of bio-inspired IoT and underlying bio-inspired algorithms that make the ecosystem comprehensive.

### 3.1. The Architecture of Bio-Inspired IoT

Bio-inspired IoT refers to the application of principles and concepts from biological systems to design and implement IoT architectures [26]. It aims to leverage the efficiency, resilience, and adaptability observed in natural systems to enhance the performance and capabilities of IoT networks. The standard architecture of a typical IoT solution consists of three levels; however, some researchers [32,33,34,35,36] have argued that it really has a four-layer design as aforementioned [1,3,4]. While there is no specific standardized architecture for bio-inspired IoT, their architecture varies depending on the specific application and the level of bio-inspiration incorporated. However, there are certain common architectural elements that are often present in bio-inspired IoT systems. In general, the architecture of a typical bio-inspired IoT solution can be apportioned into seven layers: the physical layer, network/communication layer, data-processing/decision-making layer, adaptation and learning layer, control layer, and application layer, depicted in Figure 7.

Sensing layer

The sensing layer consists of a collection of sensors and actuators deployed in the physical environment. These sensors capture data from the surrounding environment, such as temperature, humidity, light intensity, or motion, mimicking the sensory organs of biological organisms [34,35,36,37,38].

2.Communication layer

The communication layer facilitates the exchange of data between IoT sensing devices and the further layers in the hierarchy [31,32,33,34]. It includes wireless communication protocols and networking infrastructure for seamless connectivity. On the other hand, bio-inspired communication mechanisms are also employed to optimize energy consumption, enhance reliability, or adapt to dynamic network conditions [34,35,36,37,38].

3.Data-processing layer

The data-processing layer is responsible for analyzing and processing the data collected by the sensors. This layer may include bio-inspired algorithms for data fusion, pattern recognition, machine learning, or optimization, drawing inspiration from biological processes like neural networks, genetic algorithms, or swarm intelligence [34,35,36,37,38].

4.Decision-making layer

The decision-making layer incorporates bio-inspired mechanisms to enable autonomous and decentralized decision making. This layer can involve algorithms inspired by collective intelligence, such as swarm intelligence, where individual IoT devices interact and cooperate to make decisions [34,35,36,37,38].

5.Adaptation and learning layer

The adaptation and learning layer enables IoT devices to adapt their behavior and learn from their environment. It may include AI algorithms that enable devices to learn from data, adjust their parameters, or optimize their performance over time, similar to the adaptive behavior observed in biological systems [33,34,35,36,37,38].

6.Control layer

The control layer coordinates and manages the operations of the IoT system. It may involve hierarchical control mechanisms inspired by the organization and coordination found in biological systems, allowing for self-organization, self-healing, and fault tolerance [34,35,36,37,38].

7.Application layer

The application layer encompasses the specific applications and services enabled by bio-inspired IoT systems used in various domains such as agriculture, smart city, and so on as aforementioned.

It is noted that the architecture can be tailored to the specific requirements of the application. On the other hand, the level of bio-inspiration can vary, ranging from incorporating bio-inspired algorithms within specific layers to developing fully bio-inspired architectures [34,35,36,37,38]. Whilst developing bio-inspired IoT solutions, the following key principles are also incorporated into the architecture towards making sure such solutions can overcome the challenges associated with traditional IoT systems.

Self-organization

Bio-inspired IoT architectures emphasize self-organization, allowing devices to autonomously form networks, adapt to changing conditions, and dynamically reconfigure themselves [10,11,12,13]. This concept draws inspiration from biological systems, such as ant colonies or flocking birds, where individual entities collectively organize and collaborate [26].

Swarm intelligence

Bio-inspired IoT can leverage swarm intelligence, which is inspired by the collective behavior of social insects like bees or ants. In this approach, IoT devices interact and share information to achieve specific goals through distributed decision making and coordination. Swarm intelligence helps especially in achieving scalability, fault tolerance, and robustness in IoT networks [20,21,22,23].

Hierarchical structures

Inspired by biological systems with hierarchical structures, bio-inspired IoT architectures often adopt layered or hierarchical arrangements. This allows for efficient data processing and decision making at different levels, enabling distributed intelligence and reducing the need for centralized control [15,16,17,18,19].

Adaptive and resilient

Bio-inspired IoT architectures aim to be adaptive and resilient, capable of responding to changes and disruptions. They can dynamically adapt to environmental conditions, reconfigure themselves in the presence of failures or changes, and continue operating reliably [24,25,26].

Energy efficiency

Energy efficiency is a critical consideration in IoT deployments. Bio-inspired approaches can help optimize energy consumption by taking cues from energy-efficient mechanisms observed in biological systems, such as efficient communication, resource allocation, and power management strategies [24,25,26].

Sensing and actuation

Bio-inspired IoT architectures often emphasize advanced sensing capabilities, drawing inspiration from biological sensors and perception mechanisms [5,6,7,8,26]. These architectures may also incorporate actuation mechanisms to enable devices to interact with the physical world, much like organisms found in nature.

### 3.2. Bio-Inspired IoT Algorithms

Having described the core part of bio-inspired IoT, this section discusses further bio-inspired IoT algorithms. Bio-inspired IoT algorithms refer to the use of nature-inspired approaches to develop efficient and adaptive algorithms for IoT systems [3,5,24,25,26,27,28]. These algorithms are based on the principles of biology and are designed to solve complex problems more efficiently and effectively [39,40]. Overall, bio-inspired optimization algorithms are an emerging approach that is based on the principles of and inspiration from the biological evolution of nature to develop new and robust computing techniques. There are various taxonomies available for bio-inspired algorithms [18,24,25], each with its own pros and cons, and the following Figure 8 depicts a possible classification of bio-inspired IoT algorithms.

Physically Inspired Algorithms (PIAs)

In general, PIAs draw inspiration from physical principles and phenomena to solve complex problems. These algorithms attempt to replicate or mimic physical processes or behaviors to develop efficient and effective problem-solving strategies. By harnessing the principles observed in the physical world, PIAs offer alternative approaches to traditional computational algorithms [25]. e.g., simulated annealing, harmony search algorithm. 

2.Neural Networks and Artificial Neural Networks (ANNs)

These algorithms aim to simulate the structure and functionality of biological neural networks, such as the human brain. In general, they are computational models composed of interconnected artificial neurons, which process and transmit information to perform tasks like pattern recognition, classification, and prediction [40,41,42,43,44,45]. e.g., feedforward neural networks, perceptron, convolutional neural networks. 

3.Evolutionary Algorithms (EAs)

EAs are a class of computational optimization algorithms inspired by the principles of biological evolution and natural selection. These algorithms mimic the process of natural evolution to search for optimal solutions to complex problems [45,46,47]. EAs operate on a population of candidate solutions, evolving and improving them over successive generations through the application of genetic operators such as mutation, crossover, and selection. e.g., genetic algorithms. 

4.Immunological Algorithms (IAs)

IAs are inspired by the principles of the immune system. These algorithms draw analogies from the behavior, mechanisms, and processes of the immune system to solve complex optimization and pattern recognition problems. IA models aim to mimic the adaptive and self-organizing nature of the immune system to develop intelligent algorithms [25,26,27]. e.g., artificial immune systems, clonal selection algorithm, immune network algorithm. 

5.Swarm Intelligence Algorithms (SIAs)

SIAs are a class of computational methods inspired by the collective behavior of social insects, such as ant colonies and bee swarms, or bird flocks [6,9,27]. These algorithms mimic the decentralized decision making and self-organization observed in nature, where individual agents interact locally with their environment and with each other to achieve a collective goal [27].

SIAs have found applications in various fields, including optimization, robotics, data clustering, and IoT. In the context of IoT, SIAs can be leveraged to address several challenges related to scalability, adaptability, and fault tolerance. These algorithms enable IoT devices to cooperate and coordinate their actions to perform complex tasks and solve problems efficiently [6,27,48,49,50]. One popular swarm intelligence algorithm used in IoT is Ant Colony Optimization (ACO). ACO is inspired by the foraging behavior of ants and has been applied to solve optimization problems in IoT networks. ACO works by simulating the behavior of ants searching for the shortest path between their colony and a food source. The pheromone trails left by ants guide the other ants in their search, gradually converging toward the optimal solution [27,48,49,50]. In the context of IoT, ACO can be used to optimize routing paths, resource allocation, or energy management in a network of interconnected devices [48,49,50]. Another swarm intelligence algorithm is Particle Swarm Optimization (PSO), which is inspired by the flocking behavior of birds, where each individual (particle) adjusts its position and velocity based on its own experience and the experience of the best-performing particle in the swarm. PSO has been successfully applied to various IoT applications, including dynamic task scheduling, load balancing, and sensor placement optimization. These are just a few examples of swarm intelligence algorithms used in the construction of IoT solutions. Other swarm intelligence algorithms like Genetic Algorithms (GAs), Bacterial Foraging Optimization (BFO), and the Firefly Algorithm (FA) have also been explored in the context of IoT [27,48,49,50].

6.In addition to the bio-inspired algorithms mentioned earlier, there are several other notable bio-inspired algorithms. Here are a few more examples:
A.Bat Algorithm (BA)The BA takes inspiration from the echolocation behavior of bats. It simulates the movement and interaction of bats to search for optimal solutions. Bats emit ultrasonic sounds and use the echo to detect objects and navigate their environment. The algorithm incorporates this behavior to optimize problem solutions [27,48,49,50].B.Cuckoo Search (CS)The CS algorithm is inspired by the brood parasitism behavior of cuckoos. Cuckoos lay their eggs in the nests of other bird species, and the algorithm mimics this behavior to optimize solutions. It uses random walk and Lévy flights to explore the search space and replace poor solutions with better ones [27,48,49,50].C.Bee Algorithm (BA)The BA is inspired by the foraging behavior of honeybees. It mimics the process of food source exploration and exploitation by a bee colony to solve optimization problems [27,48,49,50]. The algorithm employs employed bees, onlooker bees, and scout bees to search for high-quality solutions.D.Grey Wolf Optimizer (GWO)The GWO is inspired by the social hierarchy and hunting behavior of grey wolves. It imitates the leadership and cooperation of wolf packs to optimize solutions. The algorithm defines four types of wolves (alpha, beta, delta, and omega) and simulates their search for prey to find optimal solutions [27].E.Dolphin Echolocation Algorithm (DEA)The DEA is inspired by the echolocation behavior of dolphins. It mimics the use of sound waves and echoes for navigation and prey detection. The algorithm employs the concept of wavefronts and sonar sensing to optimize problem solutions [27].


According to our analysis, bio-inspired IoT algorithms offer several advantages over traditional algorithms when it comes to solving complex problems pertaining to IoT systems. The following describes the benefits of these bio-inspired IoT algorithms:Adaptability

Bio-inspired algorithms are inherently adaptable and can dynamically adjust to changing conditions. They draw inspiration from biological systems that exhibit adaptability and can handle uncertainties and fluctuations effectively [23,24,25]. This adaptability allows bio-inspired IoT algorithms to respond to varying network conditions, resource availability, or environmental changes, resulting in improved system performance.

2.Scalability

Bio-inspired algorithms, such as swarm intelligence, are well-suited for large-scale IoT systems [27]. They enable devices to interact and cooperate locally, leading to self-organization and decentralized decision making. This scalability feature allows bio-inspired IoT algorithms to efficiently handle a growing number of devices, interactions, and data in complex IoT networks.

3.Optimization and efficiency

Bio-inspired algorithms excel at optimization tasks, resource allocation, and decision making. By mimicking natural processes like genetic evolution or ant colony foraging behavior, bio-inspired IoT algorithms can optimize routing, scheduling, energy management, and other resource-intensive tasks [48,49,50,51,52]. This optimization leads to improved efficiency, reduced energy consumption, and enhanced system performance [53,54,55,56,57].

4.Robustness and resilience

Bio-inspired IoT algorithms often exhibit robustness and resilience similar to biological systems. They can handle failures, disruptions, or changes in the system, thanks to their decentralized and self-organizing nature. This resilience helps bio-inspired IoT systems maintain functionality and adapt to evolving conditions, making them more reliable in dynamic and uncertain environments [48,49,50,51,52,58,59,60,61,62].

5.Distributed intelligence

Bio-inspired IoT algorithms enable distributed intelligence and decision making. Instead of relying on a centralized authority, devices in bio-inspired IoT systems interact locally and collectively contribute to the decision-making process. This distributed intelligence enhances fault tolerance, autonomy, and scalability while reducing dependence on a single point of failure [48,49,50,51,52,63,64,65].

6.Sustainability and energy efficiency

Bio-inspired algorithms often prioritize energy efficiency and resource optimization, aligning with sustainability goals. They leverage energy-efficient behaviors observed in biological systems, helping reduce energy consumption and environmental impact in IoT deployments [6,13,14,15,16,64,65,66,67,68].

Overall, bio-inspired IoT algorithms offer a new approach to developing efficient and adaptive algorithms for IoT systems. As this field continues to grow, we can expect to see more innovative and efficient algorithms that are inspired by nature.

## 4. A Summarization of Related Work

In recent years, there has been a clear growth of research studies seen in the area of bio-inspired IoT. Thus, to provide a better understanding, Table 2 summarizes the latest research in terms of the domain it is applied to, the key applications it is going to optimize, its main contributions, and the scope of the work.

With the summarization of the literature we have analyzed, it is evident that bio-inspired IoT is applied to almost all the domains we have mentioned above, such as smart cities, transportation, agriculture, and so on. Further, in terms of the specifical applications it is going to optimize, we have noted several applications such as security, routing, energy consumption, path planning, anomaly detection, performance, and resource allocation.

### 4.1. Comparison of Recent Survey Studies

The following Table 3 provides a brief summary and a comparison of recent survey studies with our work. The survey articles presented in the summary were chosen based on the scope of the survey, whether the authors have discussed bio-inspiration, bio-inspired algorithms, or bio-inspired IoT, and its latest status and anticipated future directions.

Overall, in recent years there has been clear growth in the number of research activities in the context of bio-inspired IoT whereas no reviews/surveys have been conducted specifically on bio-inspired IoT, owing to the novelty of the field. Thus, with our review of bio-inspired IoT, we believe our work would inspire future research in the context of bio-inspired IoT.

### 4.2. Comparison of IoT and Bio-Inspired IoT

Based on the state of the art, the following Table 4 highlights key differences between traditional IoT and bio-inspired IoT. Accordingly, it is evident that bio-inspired IoT is far ahead of traditional IoT as bio-inspired IoT is capable of overcoming most of the constraints pertaining to traditional IoT.

## 5. Challenges Pertaining to Bio-Inspired IoT

While bio-inspired IoT offers varying benefits as opposed to traditional IoT, it also faces certain challenges. The following describes key challenges pertaining to bio-inspired IoT as we have noted through the state of the art.

Complexity and scalability

Bio-inspired algorithms often involve complex interactions and computations, especially when simulating swarm intelligence or evolutionary processes. Scaling up these algorithms to handle large-scale IoT systems with a massive number of devices and data points can be a challenging task. Thus, efficiently managing the complexity and ensuring the scalability of bio-inspired IoT solutions require careful design and optimization [34,35,36].

2.Resource constraint nature

IoT devices are typically resource-constrained in terms of computational power, memory, energy, and bandwidth [32]. Bio-inspired algorithms, especially those involving complex computations or large populations, can impose significant resource demands on IoT devices. Thus, designing bio-inspired IoT solutions that are efficient in terms of computational requirements and resource usage becomes crucial to overcome these constraints [32,34,35,36].

3.Adaptability and robustness

Bio-inspired algorithms are known for their adaptability and robustness. However, adapting these algorithms to dynamic and evolving IoT environments poses challenges. IoT systems experience changes in device availability, network conditions, and data characteristics, requiring bio-inspired IoT solutions to be resilient and adaptive to such variations [27,35,49].

4.Security and privacy

In general, IoT systems raise concerns about security and privacy, and bio-inspired IoT is no exception. The interconnected nature of IoT devices and the use of bio-inspired algorithms can introduce vulnerabilities, such as attacks on swarm intelligence or genetic processes [73]. Thus, ensuring the security and privacy of data, communications, and algorithms in bio-inspired IoT is an ongoing challenge that many researchers are currently working on [34,35,36].

5.Interoperability and standardization

Bio-inspired IoT solutions may involve different devices, protocols, or platforms, leading to interoperability challenges [32,33,34,35,36]. Integrating diverse devices, algorithms, and data formats can be complex, requiring standardized interfaces and protocols for seamless communication and collaboration. Hence, developing interoperable bio-inspired IoT solutions is crucial for their widespread adoption and integration into existing IoT ecosystems.

6.Ethical and legal considerations

Bio-inspired IoT solutions may raise ethical and legal concerns, particularly when they involve autonomous decision making, learning from data, or privacy-sensitive information. Addressing issues related to accountability, fairness, transparency, data privacy, and complying with available policies and regulations is essential to ensure responsible deployment and use of bio-inspired IoT solutions.

In summary, addressing all these challenges requires interdisciplinary research, collaboration, and a holistic approach that considers the technical, computational, environmental, social, and ethical aspects of bio-inspired IoT systems.

## 6. Current Status and Future Directions

Bio-inspired IoT aims to address various challenges in traditional IoT systems, such as scalability, energy efficiency, adaptability, and fault tolerance. Drawing inspiration from biological systems, it aids in designing IoT systems that can self-organize, self-heal, and adapt to changing environments. As of now, according to the latest statistics, there are around 16 billion IoT devices, a number which is expected to double by 2030 [29,30,31]. On the other hand, the market size was valued at nearly USD 600 billion in 2022, and it is expected to reach more than USD 3000 billion in 2030 [29,30,31]. Thus, being a rapidly growing technological ecosystem, it is expected that the challenges that IoT may encounter may also grow in the long run. Hence even though the research in terms of bio-inspired IoT is still at an early stage, it is expected that it will also continue to grow toward designing robust, safe, and resilient IoT solutions in the long run [74,75]. According to our review, it is evident that the integration of bio-inspired solutions with the IoT holds tremendous potential for shaping the future of technology and determining the future of the human race, enabling us to overcome whatever challenges we encounter. Such bio-inspired IoT solutions would continue to grow in the long run, and in this regard, anticipated future directions of bio-inspired IoT are highlighted in the following for better understanding.

Hybrid approaches

The integration of bio-inspired algorithms with other computational techniques, such as machine learning, deep learning, and data analytics, can enhance the capabilities of bio-inspired IoT [63,64,73,74,75,76,77,78]. Hybrid approaches that combine the strengths of different algorithms and methodologies are likely to emerge, enabling more powerful and adaptable solutions [74,75,79].

2.Integration with edge computing/fog computing

With the growing emphasis on edge /fog computing in IoT, bio-inspired algorithms can be leveraged at the edge to enable intelligent decision making and resource optimization [30,31,32,33,34]. Bio-inspired edge intelligence can lead to more autonomous, energy-efficient, and responsive IoT systems that process data and make decisions locally, reducing latency and bandwidth requirements.

3.Explainability and interpretability

As bio-inspired IoT systems become more complex and autonomous, the need for explainable and interpretable models and algorithms becomes critical. Research focusing on understanding and interpreting the decision-making processes of bio-inspired algorithms can lead to more trustworthy and transparent IoT systems, enabling users to comprehend and validate the outputs and behavior of such systems [48,49,50,73,74,75].

4.Collaborative and cooperative bio-inspired IoT systems

Exploring the collective behavior and cooperation among multiple bio-inspired IoT systems can lead to innovative solutions [76,77]. Research on collaborative algorithms, coordination mechanisms, and collective decision making can enable IoT systems to work together as intelligent collectives, sharing information and optimizing tasks in a cooperative manner [73].

5.Ethical and social considerations

As bio-inspired IoT becomes more integrated into various aspects of daily life, addressing ethical and social implications is crucial. Hence, future research should focus on understanding and addressing issues related to bias, fairness, accountability, transparency, and the social impact of bio-inspired IoT systems [49,73,74,75].

## 7. Conclusions

With the emergence of bio-inspired IoT, researchers have been able to develop creative solutions to handle the complex issues faced by IoT deployments by pulling inspiration from biological systems such as diverse ecosystems and swarms. In the study, we have carried out a review of what is meant by bio-inspiration, how bio-inspired IoT can overcome the challenges associated with traditional IoT, bio-inspired solutions, and the ecosystem of bio-inspired IoT, highlighting challenges and future directions. With the review performed, it is evident that bio-inspired IoT has shown exceptional flexibility and robustness in various domains where traditional IoT solutions are not capable of delivering such benefits. According to the state of the art, researchers have suggested decentralized and autonomous designs for IoT networks based on the self-organizing and self-healing qualities of natural systems, allowing them to dynamically adapt to changing conditions, reorganize themselves, and recover from failures. These bio-inspired techniques have the potential to provide extremely scalable and durable IoT systems that can survive disturbances while maintaining consistent connections. Nonetheless, it is also evident that the bio-inspired IoT paradigm has considerable environmental advantages where researchers have created energy-efficient communication protocols, low-power sensor networks, and intelligent data-processing approaches by mimicking nature’s efficiency and sustainability. These developments not only reduce the carbon footprint of IoT installations but also allow IoT technologies to be integrated into environmentally benign applications such as smart agriculture, wildlife monitoring, and smart cities. However, while bio-inspired IoT systems have enormous promise, there are several difficulties and prospects for further study. The scalability and security of these systems continue to be significant challenges, similar to traditional IoT solutions, necessitating more research in the area. On the other hand, ethical problems, privacy concerns, and the responsible use of data must all be properly addressed for bio-inspired IoT to be a stable technology, which researchers are currently working on. In conclusion, bio-inspired IoT solutions will surely play a critical part in defining the next generation of smart and pervasive solutions enhancing the power of the biological world, ushering us into a more efficient, robust, and harmonious future as we continue to investigate the synergy between nature and technology.

## Figures and Tables

**Figure 1 biomimetics-08-00373-f001:**
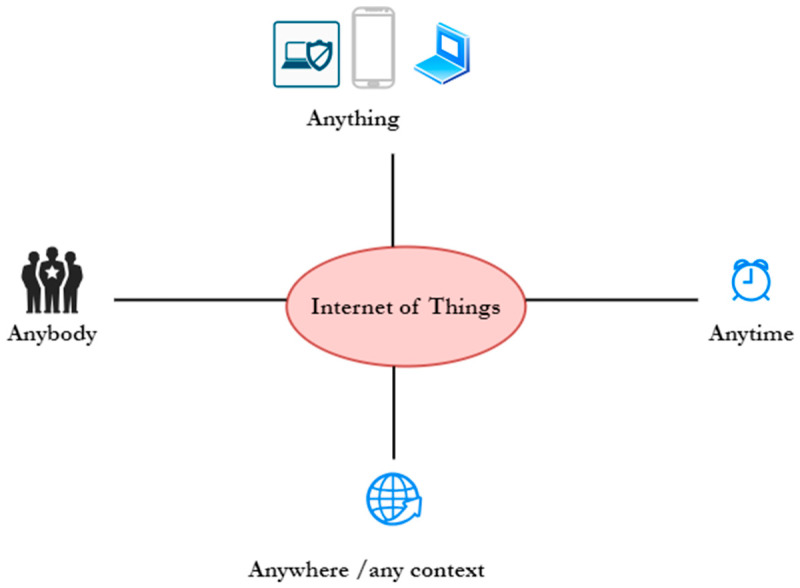
The IoT ecosystem.

**Figure 2 biomimetics-08-00373-f002:**
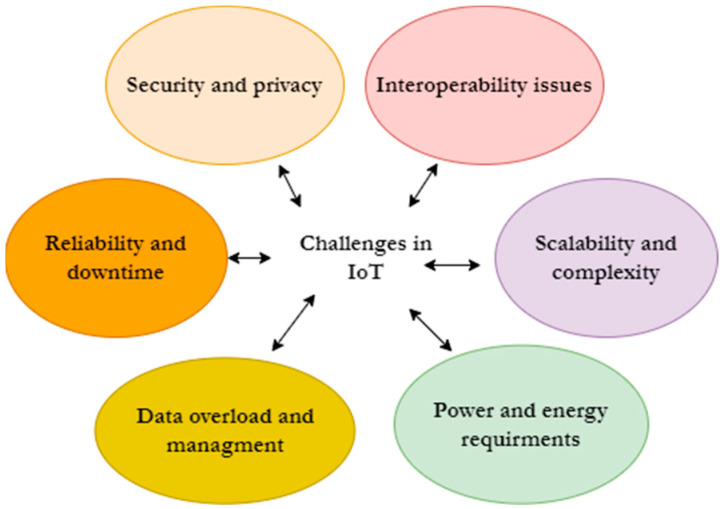
Challenges pertaining to IoT.

**Figure 3 biomimetics-08-00373-f003:**
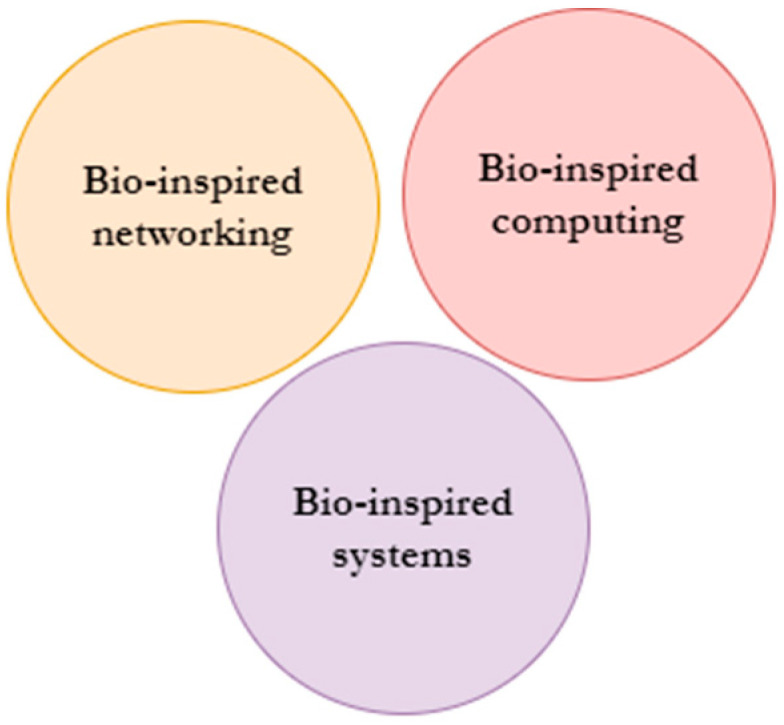
Fields of bio-inspired study.

**Figure 4 biomimetics-08-00373-f004:**
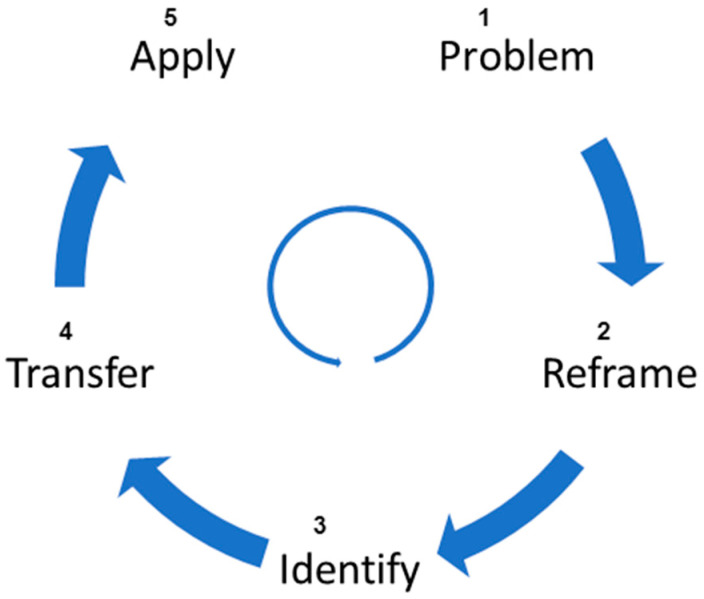
Process of designing a bio-inspired solution.

**Figure 5 biomimetics-08-00373-f005:**
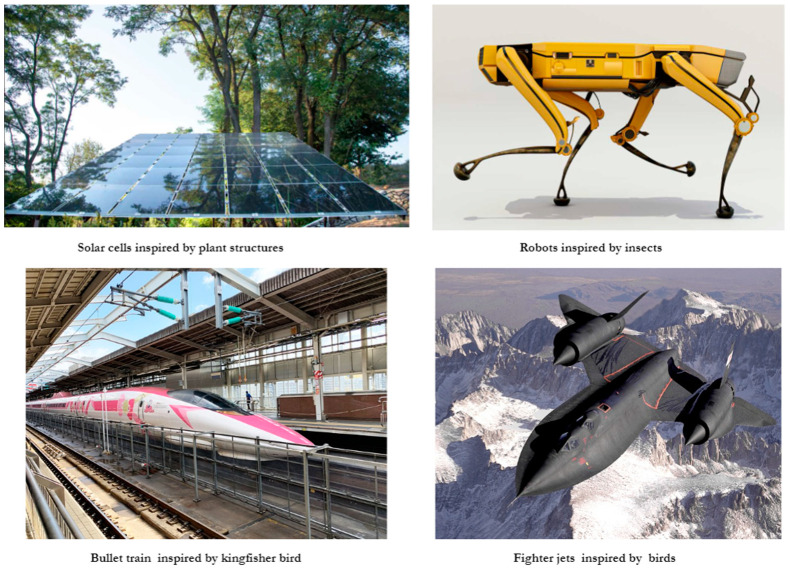
Bio-inspired solutions that have been introduced in recent years.

**Figure 6 biomimetics-08-00373-f006:**
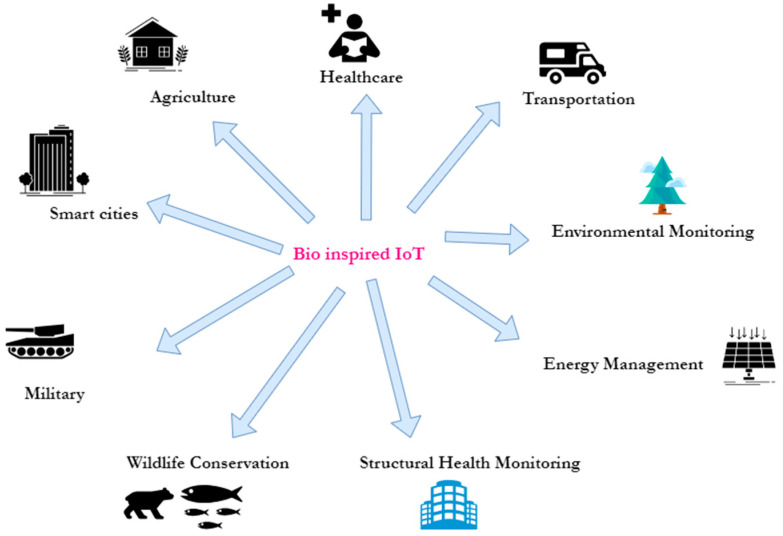
Domains of bio-inspired IoT.

**Figure 7 biomimetics-08-00373-f007:**
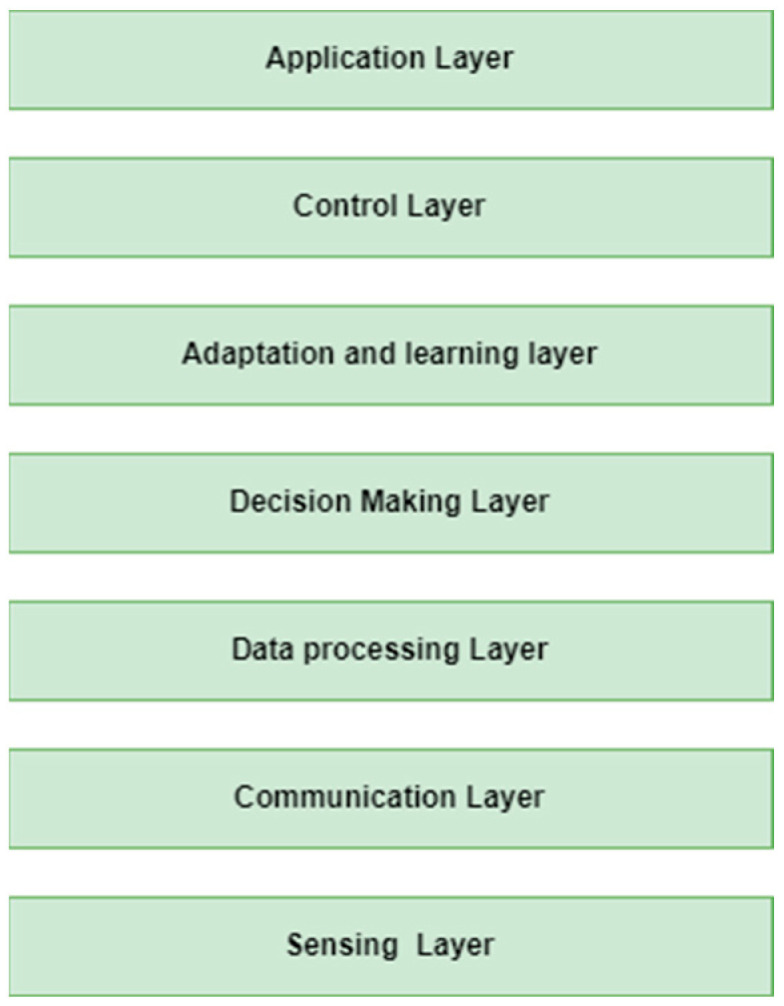
The typical architecture of bio-inspired IoT solution.

**Figure 8 biomimetics-08-00373-f008:**
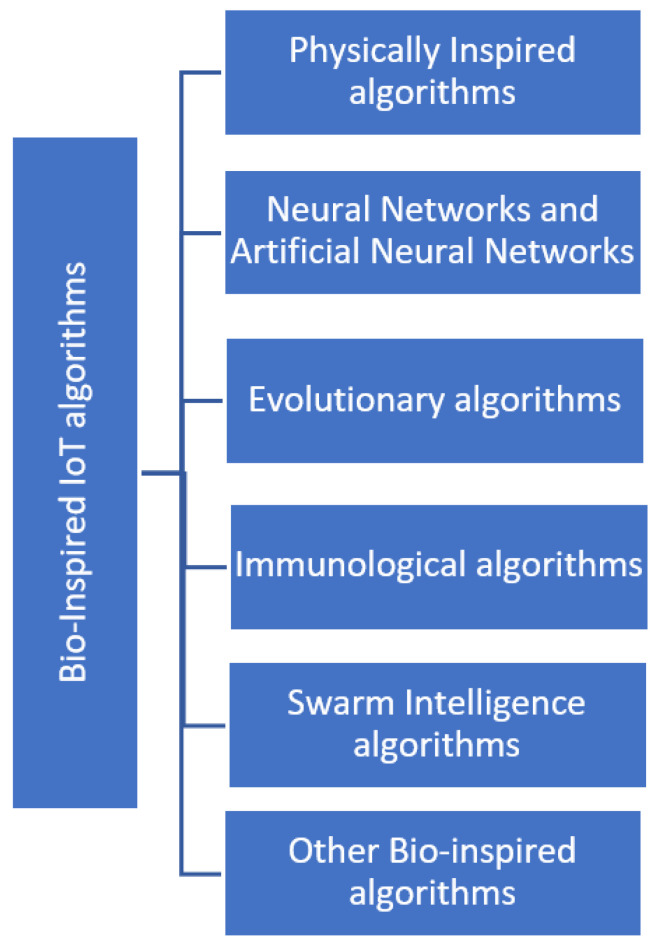
Taxonomy of bio-inspired algorithms.

**Table 1 biomimetics-08-00373-t001:** Acronyms found in this paper.

Acronym	Description
IoT	Internet of Things
ICT	Information and Communication Technology
GA	Genetic Algorithm
ACO	Ant Colony Optimization
AI	Artificial Intelligence
ANN	Artificial Neural Network
EA	Evolutionary Algorithm
IA	Immunological Algorithm
SIA	Swarm Intelligence Algorithm
PSO	Particle Swarm Optimization
BFO	Bacterial Foraging Optimization
FA	Firefly Algorithm
BA	Bat Algorithm
CS	Cuckoo Search
BA	Bee Algorithm
GWO	Grey Wolf Optimizer
DEA	Dolphin Echolocation Algorithm
HIOA	Hybrid Intelligent Optimization Algorithm
SCA	Sine–Cosine Algorithm
SSA	Salp Swarm Algorithm
ANTPSOAODV	ANT Particle Swarm Optimization Adhoc On-demand Distance Vector
BiHCLR	Bio-inspired Cross-Layer Routing
WSN	Wireless Sensor Network
MPSO	Modified Particle Swarm Optimization
MCSO	Modified Cat Swarm Optimization
UAV	Unmanned Aerial Vehicle
MOO	Multi-Objective Optimization
PSGWO	Particle Swarm Grey Wolf Optimization
BSCA	Bio-Inspired Self-Learning Coevolutionary Algorithm
GWO	Grey Wolf Optimizer
WOA	Whale Optimization Algorithm
ICA	Imperialist Competitive Algorithm
CH	Cluster Head
GRN	Gene Regulatory Network
BiO4SeL	Bio-Inspired Optimization for Sensor Network Lifetime
BIOSARP	Bio-Inspired Self-Organized Secure Autonomous Routing Protocol
SDAR	Secured Data Assured Routing

**Table 2 biomimetics-08-00373-t002:** Summary of latest research pertaining to bio-inspired IoT.

Reference	Domain	Application/s	Main Contribution	Scope of the Work
[1]	IoT network communication	Security	Proposes an AI-assisted bio-inspired algorithm.	The authors proposed an AI-assisted bio-inspired algorithm for securing IoT communication. Their proposed framework comprises two components. One component includes bio-inspired algorithm-assisted blockchain technology for authentication and authorization, whereas the other component includes an AI algorithm that keeps an eye on the IoT communication network.
[2]	Manufacturing	Scalability	Introduces a novel bio-inspired control architecture for modern cyber-physical manufacturing systems.	This research introduces a novel bio-inspired control architecture for modern manufacturing systems, which suggests an IoT-enabled framework for detecting motor abnormalities using vibration sensors. The proposed approach employs a real-time autoencoder for enhanced accuracy. In contrast to existing methodologies, this study focuses on analyzing the behavior of anomaly detection in real time.
[3]	Cyber-physical systems	Security	Presents a bio-inspired method for the identification of hardware trojans.	The authors present a bio-inspired method for the identification of hardware trojans in cyber-physical systems. Further, they also developed a bio-inspired device-locking mechanism, which they used in order to construct a design-for-trust architecture. The findings proved that the concept is suitable for resource-constrained situations that have low hardware and power dissipation profiles.
[4]	IoT network communication	Security	Presents a novel bio-inspired approach to enhance the security of distributed IoT devices.	The authors present a novel approach inspired by biology to enhance the security of distributed IoT devices. The main objective of their proposed framework is to identify, refuse, and prevent unauthorized external agents from accessing the devices, both individually and in cooperation, in real time.
[6]	IoT network communication	Energy consumption	Presents a novel clustering approach enabled by the combination of bio-inspired algorithms toward optimizing energy consumption.	The authors have used fuzzy logic, chicken swarm optimization, and a genetic algorithm to present an optimal cluster formation as a Hybrid Intelligent Optimization Algorithm (HIOA) to minimize overall energy consumption in an IoT network.
[7]	IoT network communication	Network slicing	A novel bio-inspired wireless resource allocation approach is introduced.	A novel wireless resource allocation approach with slice characteristic perception has been presented by the authors for use in 5G-enabled IoT networks.
[8]	IoT network communication	Routing performance	A bio-inspired decentralized service discovery and selection model is introduced.	Using the bio-inspired response threshold model as inspiration, this paper proposes a decentralized service discovery and selection model. Obtained results indicated that the proposed method exhibits efficient routing and scalability for IoT networks.
[9]	IoT network communication	Security	A novel hybridized bio-inspired intrusion detection system is introduced.	The researchers present an innovative approach for enhancing the security of the IoT framework through a hybridized bio-inspired intrusion detection system. This system utilizes a combination of two bio-inspired algorithms, namely the Sine–Cosine Algorithm (SCA) and the Salp Swarm Algorithm (SSA), to effectively analyze and identify essential network traffic patterns. By extracting relevant features, these characteristics are then forwarded to a machine learning classifier, enabling the system to accurately detect and classify intrusive traffic.
[10]	IoT network communication	Security	A novel trust-based safe data aggregation approach and an energy-efficient safe routing protocol are introduced.	Researchers propose ANT Particle Swarm Optimization Ad hoc On-demand Distance Vector (ANTPSOAODV), a trust-based safe data aggregation approach, and an energy-efficient safe routing protocol for a multi-hop environment in an IoT-enabled wireless sensor network.
[11]	IoT network communication	Resource utilization	A whale-based sensor clustering model is introduced.	The authors have proposed a novel distributed model to effectively manage heterogeneous sensors and select accurate ones in a dynamic IoT environment, using a bio-inspired clustering algorithm: whale-based sensor clustering.
[12]	IoT network communication	Routing and topology maintenance	A novel bio-inspired clustering algorithm is proposed.	The authors have proposed a novel bio-inspired clustering algorithm based on a honeybee algorithm, genetic algorithm, and tabu search for IoT-enabled mobile ad hoc networks.
[13]	IoT network communication	Routing	A new bio-inspired cross-layer routing protocol is purposed.	The researchers propose a new Bio-Inspired Cross-Layer Routing (BiHCLR) protocol for efficient and energy-efficient routing in IoT-enabled wireless sensor networks.
[14]	IoT network communication	Security	A novel bio-inspired secure IPv6 communication protocol is proposed.	The researchers propose a novel bio-inspired secure Ipv6 communication protocol for the IoT.
[15]	IoT network communication	Energy consumption	A new fusion cluster head selection technique is proposed.	To maximize the amount of time that a wireless sensor network is operational, the optimal selection of cluster heads is a crucial criterion that must be met. With this in mind, the authors present a new fusion cluster head selection technique that combines the advantages of the LEACH protocol and the dragonfly algorithm.
[16]	Agriculture	Energy consumption and operational time	A bio-inspired self-learning coevolutionary algorithm is presented.	The authors present a bio-inspired self-learning coevolutionary algorithm for dynamic multi-objective optimization of IoT services to cut down on energy usage and service time.
[17]	IoT network communication	Routing	A bio-inspired intelligent routing schema is proposed.	To reduce the amount of energy an IoT network consumes, an intelligent routing scheme based on a bio-inspired technique is proposed that can significantly extend the IoT network’s lifetime.
[18]	IoT network communication	Resource allocation	A new bio-inspired algorithm is presented for distributed resource allocation.	The authors introduced a multi-hop DESYNC algorithm, which is a bio-inspired Time Division Multiple Access (TDMA)-based strategy for distributed resource allocation in sensor networks. The DESYNC algorithm draws inspiration from biological systems to allocate distributed resources efficiently.
[19]	IoT network communication	Routing	A customized queen honeybee migration algorithm is presented.	The researchers have enhanced the original queen honeybee migration algorithm, which was introduced for efficient mobile routing in WSN, using binary testing injection on the cooperative node’s selection on the IoT system.
[20]	Fog computing	Resource management	A new bio-inspired algorithm is presented for resource allocation.	The authors suggested a new bio-inspired hybrid algorithm, which they referred to as the NBI-HA. This approach is a cross between Modified Particle Swarm Optimization (MPSO) and Modified Cat Swarm Optimization (MCSO). The hybrid of the MPSO and MCSO is utilized to manage resources at the fog device level in the proposed method.
[21]	Edge computing	Security	A novel bio-inspired approach is presented for enhancing the security of IoT applications.	The authors have demonstrated a combination of IoT peripheral sensors and low-power crypto engines. Using bio-inspired systems as inspiration, Two-Dimensional (2D) memtransistors accomplish the integration. This “all-in-one” solution seeks to enhance the functionality and security of IoT applications.
[24]	IoT network communication	Performance and energy consumption	A novel bio-inspired technique is presented in conjunction with fuzzy logic.	The authors present a technique that integrates fuzzy logic with various nature-inspired algorithms—grey wolf algorithm and firefly algorithm—to effectively balance the burden among IoT devices in a network.
[37]	IoT network communication	Resource allocation	A new firefly-based clustering approach is presented.	The authors propose a new firefly-based clustering approach for IoT applications.
[38]	Unmanned Aerial Vehicles (UAVs)	Optimal path planning	A bio-inspired optimal path planning schema is presented.	Using a joint genetic algorithm and ant colony optimization, the authors have proposed an optimal flight planning schema for UAVs.
[39]	IoT network communication	Energy consumption	A novel energy-aware clustering schema is presented.	Inspired by Particle Swarm Optimization (PSO), the authors propose a novel energy-aware bio-inspired clustering scheme (PSO-WZ) for IoT network communication.
[40]	IoT network communication	Energy consumption	A novel bio-inspired energy optimization approach is presented.	The authors propose a novel Multi-Objective Optimization (MOO) agent based on Particle Swarm Grey Wolf Optimization (PSGWO) and inverse fuzzy ranking for energy optimization of IoT networks.
[43]	Smart vehicles	Energy consumption	A bio-inspired smart vehicle design is presented.	The researchers have designed a bio-inspired smart vehicle with an AI-enabled charging system.
[44]	IoT network communication	Data exchange	A novel bio-inspired approach is presented for data exchange over WSNs.	Two algorithms, Grey Wolf Optimizer (GWO) and Whale Optimization Algorithm (WOA), in conjunction with the Imperialist Competitive Algorithm (ICA)-based Cluster Head (CH) selection and a novel approach, are proposed for heterogeneous networks. These algorithms facilitate data exchange over heterogeneous WSN infrastructures by addressing the buffer overflow issue.
[46]	Smart city	Energy consumption and quality of data	A novel bio-inspired distributed event-sensing and data collection framework is presented.	Based on Gene Regulatory Networks (GRNs) in living organisms, this paper proposes bioSmartSense, a novel bio-inspired distributed event-sensing and data collection framework. The objective is to make the sensing and reporting processes more energy efficient.
[47]	Fog computing	Service allocation	A novel bio-inspired algorithm is presented for service allocation.	The researchers have developed a hybrid algorithm using a genetic algorithm and particle swarm optimization technique to solve the service allocation problems pertaining to fog computing.
[48]	Transportation	Anomaly detection (detection of road cracks)	A bio-inspired deep learning approach is presented.	The authors have proposed an IoT system with a bio-inspired deep learning approach for accurate road crack detection.
[51]	IoT network communication	Energy consumption and routing	A novel bio-inspired routing algorithm is presented.	The researchers have presented a novel routing algorithm designed to extend the longevity of the network and conserve the energy of sensor nodes connected to the WSN. The proposed algorithm is a hybrid of genetic and ant colony optimization algorithms.
[52]	IoT network communication	Security	A novel bio-inspired ensemble classifier is introduced.	The authors have introduced a novel bio-inspired ensemble classifier towards improving the performance of anomaly detection of IoT networks.
[53]	IoT network communication	Security	A novel layered artificial immune system approach, inspired by the natural immunity mechanism, is proposed.	The authors have introduced a novel layered artificial immune system approach, inspired by the natural immunity mechanism, and adapted an architecture called ImmuneGAN to identify the affected network packets in the IoT network to detect security anomalies.
[54]	IoT network communication	Security	A novel secure and lightweight dynamic encryption bio-inspired model is introduced.	The researchers have introduced a design for a novel secure and lightweight dynamic encryption bio-inspired model for IoT networks and demonstrated that it applies to a broad range of low-complexity IoT deployments.
[55]	IoT network communication	Energy consumption and routing	A novel bio-inspired algorithm is proposed for determining the optimal path.	Researchers propose a novel algorithm based on ant colony optimization for determining the optimal path for data transmission in WSNs.
[56]	Healthcare	Image processing	A novel swarm intelligence-based image processing approach is introduced.	The authors have proposed a novel swarm intelligence-based approach for lung cancer detection and transmission of gathered data to the cloud.
[57]	IoT network communication	Routing	A novel bio-inspired middleware for WSN is introduced.	The authors have introduced a novel bio-inspired middleware for WSNs, with the aim of introducing self-adaptive architecture.
[58]	Fog computing / mist computing	Data distribution	A bio-inspired algorithm for data distribution is introduced.	The researchers have proposed a novel bio-inspired algorithm for data distributions in fog and mist computing environments.
[59]	IoT network communication	Security	A novel intrusion detection system inspired by the grey wolf algorithm is introduced.	The authors have introduced an intrusion detection system modeled as a two-stage framework with feature selection performed by a generalized mean grey wolf algorithm and an elastic net contractive autoencoder for classifying malicious traffic in IoT networks.
[62]	IoT network communication	Routing	A novel ant colony metaphor-based approach is introduced for optimized routing.	The researchers introduce AntNet, a novel approach to the adaptive learning of routing tables in communications networks.
[68]	UAVs	Route selection	A novel bio-inspired clustering scheme is introduced.	The researchers propose a novel bio-inspired clustering scheme using a dragonfly algorithm for cluster formation and management in route planning for UAVs.
[69]	IoT network communication	Self-organization and routing	A novel swarm intelligence-based algorithm is introduced for performing self-organization in WSNs.	The researchers present Bio-Inspired Optimization for Sensor Network Lifetime (BiO4SeL), a swarm intelligence-based algorithm, to perform self-organization and optimization of a lifetime by means of routing into a WSN.
[70]	IoT communication network	Security	A novel bio-inspired self-organized secure autonomous routing protocol is introduced.	The authors present a Bio-Inspired Self-Organized Secure Autonomous Routing Protocol (BIOSARP) and Secured Data Assured Routing (SDAR) in WSNs.
[71]	IoT network communication	Security	A novel bio-inspired self-organized secure autonomous routing protocol is introduced.	The researchers introduce a Self-Organized Secure Autonomous Routing Protocol (BIOSARP) which enhances a WSN in securing itself from abnormalities and most common WSN routing attacks.

**Table 3 biomimetics-08-00373-t003:** Comparison of recent surveys (Yes—✔, No—🗶).

Reference	Scope of the Work	Discusses Bio-Inspiration	Discusses Bio-Inspired Algorithms	Discusses Bio-Inspired IoT	The Current Status of Bio-Inspired IoT Is Discussed	Challenges and Future Directions of Bio-Inspired IoT Are Discussed
[5]	Highlighted the statistics pertaining to the use of bio-inspired solutions and traditional technologies to overcome the challenges associated with IoT.	✔	🗶	🗶	✔	🗶
[22]	Researchers have examined the biologically inspired algorithms used for solving challenges posed by different sensor mobility schemes in the context of IoT applications.	🗶	✔	🗶	🗶	🗶
[23]	Provided a review of how IoT-based AI-enabled bio-inspired solutions can be used as a remedy to fight against cyber-crimes.	✔	✔	🗶	🗶	🗶
[25]	Provided a review of bio-inspired solutions that can be used to solve complex engineering problems.	🗶	✔	🗶	🗶	🗶
[27]	Presents a comprehensive review of the state of the art, nine bio-inspired computing algorithms, and their applications.	✔	✔	🗶	🗶	🗶
[36]	Provides perspectives on bio-inspired technologies and offers a brief discussion on how such technologies can be used to solve day-to-day challenges in a low-cost and sustainable way.	✔	🗶	🗶	🗶	🗶
[41]	Explores how bio-inspired approaches can be used to secure IoT ecosystems.	🗶	✔	🗶	🗶	🗶
[45]	Examines 5G network layer security for IoT applications and provides a list of network layer security vulnerabilities and requirements for WSNs, IoT, and 5G-enabled IoT. Secondly, it provides a comprehensive review of the presented network layer security methods and bio-inspired techniques for IoT applications exchanging data packets over 5G, including analysis of bio-inspired algorithms in terms of providing a secure network layer for IoT applications connected to 5G and beyond networks.	🗶	✔	🗶	✔	✔
[49]	The authors have provided a review of bio-inspired optimization algorithms.	✔	✔	🗶	🗶	🗶
[60]	The authors have provided an in-depth analysis of swarm intelligence models and how they can be applied to complex systems.	✔	✔	🗶	🗶	🗶
[61]	Provides an in-depth insight into how swarm intelligence can be used to solve complex engineering problems.	🗶	✔	🗶	🗶	🗶
[63]	The authors discuss state-of-the-art bio-inspired research for communication technology in IoT networks.	✔	✔	🗶	🗶	🗶
[65]	The swarm intelligence algorithms in IoT are investigated with a special focus on the Internet of Medical Things.	🗶	✔	🗶	🗶	🗶
[66]	This study provides a review of swarm intelligence algorithms and their potential use in IoT-based applications.	✔	✔	🗶	🗶	🗶
[67]	The researchers provide an overview of the confluence between big data technologies and bio-inspired computation.	🗶	✔	🗶	🗶	🗶
[72]	Provides a review of swarm intelligence algorithms and summarizes their applications in the IoT.	🗶	✔	🗶	🗶	🗶
[73]	Provides a review of a set of swarm intelligence algorithms applied to the main challenges introduced by the IoT.	🗶	✔	🗶	🗶	🗶
Our work	Presents a comprehensive review on bio-inspired IoT, how it came in to play, its ecosystem, state of the art, current status, challenges, and future directions.	✔	✔	✔	✔	✔

**Table 4 biomimetics-08-00373-t004:** Difference between traditional IoT and bio-inspired IoT.

Criteria	IoT	Bio-Inspired IoT
Design	Technology-driven and focused on connectivity and data	Bio-inspired algorithms are driven by biological principles
Optimization	Centralized control and optimization algorithms	Decentralized decision making and collective intelligence
Scalability	Scalability challenges due to the increasing number of devices	Swarm intelligence allows efficient scalability
Resource efficiency	Optimization focused on energy and resource usage	Efficiency in resource allocation and energy management
Security and privacy	Traditional security and privacy concerns	Bio-inspired algorithms may introduce new security challenges
Adaptability	Limited adaptability to changing conditions	Adaptive and robust to dynamic environments similar to biological organisms
Learning and adaptation	Machine learning/deep learning techniques applied for data analysis	Bio-inspired algorithms with adaptive learning capabilities

## Data Availability

No new data is created.

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
