# Peer review of "Bio-Inspired Internet of Things: Current Status, Benefits, Challenges, and Future Directions"

_biomimetics, 2023, doi:10.3390/biomimetics8040373_

Round 1
Reviewer 1 Report
In the manuscript, the authors give a comprehensive review on the current status, benefits, challenges, and future directions of the bio-inspired IoT, which is important for the researchers in the bio-inspired study, especially for the beginners in this area. The manuscript is well-organized and written. However, some issues require to be solved before the publication.
1. The “bio inspire” should be changed to “bio-inspired” in the caption of Figure 3.
2. It is hard to understand the meaning of the direction of arrow in the middle of the Figure 4. Please give an explanation or reedit the Figure 4.
3. More examples should be given about the bio-inspired solutions in the Figure 5.
4. “Military” is overlapped with the “Smart cities” in the Figure 6. The order of the cartoons of the domains are suggested to be in accordance with the order that described in the main text.
5. The description about the application of Bio-inspired IoT in military is missing.
6. Some recent literatures on bio-inspired materials or systems should be cited, such as Exploration 2023, 20220052; Accounts of Materials Research 2021, 2, 6, 407–419; Exploration 2023, 3, 20210263; Nano Materials Science 2020, 2, 264–280.
Author Response
Dear Sir/Madam,
We have addressed all the comments provided. Kindly see the attachment.

Reviewer 2 Report
In this study, the paper briefly reviews the bioinspired Iot, focusing on how it works, its ecosystem, its latest state, benefits, challenges and future directions. Specifically, the paper reviews what is bioinspired, how bio-inspired IoT can overcome the challenges associated with traditional IoT, bioinspired solutions, and the ecosystem of bio-inspired IoT, highlighting the challenges and future directions. Bioinspired IoT shows remarkable flexibility and robustness in a variety of areas where traditional IoT solutions cannot provide such advantages. At the current state of the art, the researchers propose a decentralized and autonomous design of IoT networks based on the self-organizing and self-healing properties of natural systems, allowing them to dynamically adapt to changing conditions, self-reorganize, and recover from failures. These biologically-inspired technologies have the potential to offer highly scalable and durable IoT systems that can withstand interference while maintaining consistent connectivity. But at the same time, the biologically-inspired IoT paradigm has considerable environmental advantages, in which the researchers created energy-efficient communication protocols, low-power sensor networks, and intelligent data processing methods by mimicking the efficiency and sustainability of nature. These developments not only reduce the carbon footprint of IoT devices, but also enable IoT technologies to be integrated into environmentally friendly applications such as smart agriculture, wildlife monitoring, and smart cities. However, despite the great promise of biologically inspired IoT systems, there are some difficult and supportive aspects that require further investigation. The scalability and security of these systems still face significant challenges similar to those of traditional IoT solutions, thus requiring more research in this area.
On the other hand, ethical issues, privacy issues, and responsible use of data must all be properly addressed to make the Biological Internet of Things a stable technology that researchers are currently working on. In general, the summary made by this paper is relatively comprehensive and thoughtful and forward-looking, and we affirm the contribution of the paper.
At the same time, there are some shortcomings in the paper that need to be pointed out, I hope to be modified to make the paper more perfect.
1. The layout of the paper is slightly lacking, for example, the abbreviation table in Table 1 should be placed at the beginning of the paper will be better and convenient to consult.Please refer the recent papers: Chenggang Yan, Biao Gong, Yuxuan Wei, Yue Gao, “Deep Multi-View Enhancement Hashing for Image Retrieval”, IEEE Transactions on Pattern Analysis and Machine Intelligence, 2020.Chenggang Yan, Zhisheng Li, Yongbing Zhang, Yutao Liu, Xiangyang Ji, Yongdong Zhang, “Depth image denoising using nuclear norm and learning graph model”, ACM Transactions on Multimedia Computing Communications and Applications 2020.Chenggang Yan, Yiming Hao, Liang Li, Jian Yin, Anan Liu, Zhendong Mao, Zhenyu Chen, Xingyu Gao, “Task-Adaptive Attention for Image Captioning”, IEEE Transactions on Circuits and Systems for Video Technology, 2021. Chenggang Yan, Tong Teng, Yutao Liu, Yongbing Zhang, Haoqian Wang, Xiangyang Ji, “Precise No-Reference Image Quality Evaluation Based on Distortion Identification”, ACM Transactions on Multimedia Computing Communications and Applications 2021.Chenggang Yan, Lixuan Meng, Liang Li, Jiehua Zhang, Jian Yin, Jiyong Zhang, Zhan Wang, Bolun Zheng, “Age-Invariant Face Recognition By Multi-Feature Fusion and Decomposition with Self-Attention”, ACM Transactions on Multimedia Computing Communications and Applications 2021
2. Is there any omission when comparing the paper work with other related paper work?
3. For what is bioinspired, the introduction of bioinspired Internet of Things can be appropriately brief, highlighting the shortcomings and prospects of existing technology.
4. Figure 1 The drawing is not good, and the unclear font affects the viewing.
5. The font in Figure 2 is blurred and unclear, and the subtitle covers part of the picture.
6. The font in Figure 3 is also blurred and unclear.
7. Figure 6 The font is blurred, and some fonts overlap clearly.
8. Figure 7 The layout is a little wrong, the upper part of the image frame is blocked, and the font is also blurred.
9. Table 2 the left column references the left, some, some right, the best unified center display, look more tidy.
10. The label of the reference should be added in the middle brackets, and the line spacing is too small to affect the appearance should be adjusted to make it look neat and beautiful.
nice
Author Response

(The authors gave the same response as above.)

Reviewer 3 Report
The authors present an introduction to bio-inspired IOT and a review of the related literature.
Although the contribution remains at a rather general/superficial level and there is some unnecessary repetition, I believe it can serve well as an easy-to-understand introduction for students and researchers. I recommend that the paper be accepted for publication.
Minor comment: Please check the numbers in line 849/850
The language is fine. You might still review it and check if there are repetitions that can be avoided.
Author Response

(The authors gave the same response as above.)

Reviewer 4 Report
The manuscript focuses on an interesting research problem about bio-inspired IoT networks. Recent papers and surveys were studied.
Some recommendations to enhance the quality of the study:
1- There are some conceptual issues or lack of precision in the conceptualization of well-known terms in the paper. An example is the use of the term “Internet of Things (IoT)" and "Wireless sensor networks". The differences and similarities between the two terms should be clearly identified.
2- The content of the survey basically consists of the presentation of several techniques, described under a classification proposed by the authors, and compared with each other, through tables, in terms of main contribution and scope of work without giving the advantages and disadvantages of these works. Moreover, there is no in-depth analysis of each one, only a brief discussion summarizing the findings presented in the tables.
3- Such a comparison can be useful for researchers interested in the topic, but the lack of clear criteria and a well-defined methodology for choosing the presented works weakens the contribution.
4- The choice of compared recent survey studies is not justified. Authors should adopt a better-defined methodology for choosing the analyzed works and surveys.
Author Response

(The authors gave the same response as above.)
